

# Synergistic effects of foliar applied glycine betaine and proline in enhancing rice yield and stress resilience under salinity conditions

Sobhi F. Lamlom[1,2,*], Aly A. A. El-Banna[2], Honglei Ren[3], Bassant A. M. El-Yamany[4], Ehab A. A. Salama[5], Gawhara A. El-Sorady[2], Mohamed M. Kamara[6], Amal Mohamed AlGarawi[7], Ashraf Atef Hatamleh[7], Abdelsalam A. Shehab[8] and Ahmed M. Abdelghany[9,*]

[1] Work Station of Science and Technique for Post-doctoral in Sugar Beet Institute, Heilongjiang University, Harbin, Heilongjiang, China
[2] Plant Production Department, Faculty of Agriculture Saba Basha, Alexandria University, Alexandria, Egypt
[3] Soybean Research Institute, Heilongjiang Academy of Agricultural Sciences, Harbain, Heilongjiang, China
[4] Rice Research Department, Field Crop Research Institute, Agricultural Research Center, Alexandria, Egypt
[5] Agricultural Botany Department, Faculty of Agriculture Saba Basha, Alexandria University, Alexandria, Egypt
[6] Department of Agronomy, Faculty of Agriculture, Kafrelsheikh University, Kafrelsheikh, Egypt
[7] Department of Botany and Microbiology, College of Science, King Saud University, Riyadh, Saudi Arabia
[8] Agronomy Department, Faculty of Agriculture, Al-Azhar University, Cairo, Egypt
[9] Crop Science Department, Faculty of Agriculture, Damanhur University, Damanhur, Egypt
* These authors contributed equally to this work.

Corresponding authors
Sobhi F. Lamlom,
sobhifaid@alexu.edu.eg
Ahmed M. Abdelghany,
ahmed.abdelghany@agr.dmu.edu.eg

## ABSTRACT

Soil salinity is one of the most challenging environmental factors affecting rice productivity, particularly in regions with high saline soils such as Egypt. The ability of rice to maintain high yield and quality under saline stress is often limited, leading to significant reductions in productivity. With the increasing salinization of agricultural lands, finding effective agronomic practices and treatments to mitigate salt-induced damage in rice crops is critical for ensuring food security. This study investigates the potential of exogenous glycine betaine (GB) and proline (Pro) applications to mitigate the adverse effects of salt stress on rice (cv. Sakha 108) over two consecutive growing seasons (2021–2022). Treatments of 30 mM GB and 30 mM Pro significantly enhanced dry weight (162.2 and 169.7 g in 2021 and 2022, respectively), plant height (88.94 and 99.00 cm), tiller number (10.58 and 10.33), and grain yield (4.22 and 4.30 t/ha) compared to control groups. Combined treatments of 30 mM GB and 30 mM Pro exhibited the greatest improvements across both years, with maximum dry weight (193.44 and 186.56 g), plant height (112.00 and 112.33 cm), tiller number (15.33 and 16.28), spikelet number per meter (264.00 and 264.05), thousand-kernel weight (70.00 and 73.2 g), and grain yield (6.17 and 6.64 t/ha). Additionally, the combined treatments resulted in the highest harvest index (53.22% in 2021 and 48.94% in 2022), amylose content (24.24% and 20.09%), and protein

content (12.33% and 12.00%). Correlation analysis highlighted strong positive relationships among traits, such as plant height with grain yield (r = 0.94), biomass yield (r = 0.92), and harvest index (r = 0.90). Path analysis further demonstrated that thousand-kernel weight and biomass yield had the most significant direct effects on grain yield, with values of 0.43 and 0.42, respectively. Heatmap clustering and principal component analysis (PCA) confirmed the synergistic effects of combined GB and Pro treatments, with the 30P_30GB treatment consistently clustering with high-yield traits, enhancing nitrogen use efficiency and stress resilience. In conclusion, the combined application of glycine betaine and proline significantly enhances the agronomic and chemical traits of rice under salt stress. This study demonstrates that these osmoprotectants improve vegetative growth, grain yield, and quality, with synergistic effects observed at optimal concentrations. The findings highlight the potential of glycine betaine and proline as effective tools for improving salt tolerance in rice, offering practical solutions to address challenges in saline-affected agricultural regions.

## INTRODUCTION

Rice (*Oryza sativa* L.) is an essential cereal crop and a staple food source for more than half the world's population (*Eka, Imron & Dini, 2021*; *Jaiswal et al., 2019*). In 2020, the worldwide rice cultivation area was approximately 165.03 million hectares, with a production of 759.95 million tons (*FAO, 2023*). The global population is expected to surpass 9.6 billion by 2050 (*Hoang et al., 2016*), creating a demand for a 70% rise in food grain production to meet the increased needs (*Ganie et al., 2019*). To achieve maximum rice production, it is essential to utilize all available land resources to their fullest potential. However, this task is challenging, as it not only requires a significant increase in crop productivity but also must be achieved in a more uncertain climate. Research indicates that approximately one-third of cultivable land worldwide is affected by salt, and half of all arable land could become salinized by 2050, posing a significant threat to sustainable agriculture (*Shahid, Zaman & Heng, 2018*).

Rice is a critical staple crop in Egypt, grown on approximately 20% of the nation's cultivated land (~0.65 million ha), which positions Egypt as the leading rice producer in Africa and the Near East region (*Elbasiouny & Elbehiry, 2020*). Predominantly cultivated in the Nile Delta region, Egypt's rice fields face significant challenges due to soil salinization caused by the intrusion of saline water into agricultural lands and groundwater. This issue is compounded by the frequent use of low-quality, saline water for irrigation, impacting around 35% of agricultural land (*Elbasiouny & Elbehiry, 2020*; *Kotb et al., 2000*; *Zidan & Dawoud, 2013*). Soil salinization may reduce Egypt's rice yields by 25% by 2025 (*Elbasiouny & Elbehiry, 2020*). In response, efforts to develop high-yielding, salt-tolerant rice varieties have intensified. Sakha 108, a promising cultivar from the Sakha Research Center of Field Crops in Kafrelsheikh, Egypt, shows potential, although it is

sensitive to salt stress. Sakha 108, while also high-yielding, exhibits sensitivity to salt stress during prolonged field exposure (*Negm et al., 2023*); although the physiological and molecular bases of this sensitivity, particularly at early developmental stages, are yet to be fully understood.

Salt stress is a multifaceted abiotic stress that harms plants by inducing ionic stress due to toxic ion concentrations (mainly $Na^+$), osmotic stress due to reduced water intake, and oxidative stress due to an increase in the level of reactive oxygen species (ROS) (*Shah et al., 2021*). Such abiotic stress is predicted to impact approximately 6% of all land areas, with approximately 22% of cultivated fields and 33% of irrigated fields used for agriculture (*Patel et al., 2020*). Soil salinization is a worldwide threat because the increased level of salinization in recent years has resulted in food insecurity in numerous nations. Rising soil salt levels are linked to environmental factors like extreme temperatures and human activities such as irrigation with brackish groundwater (*Lamlom et al., 2025*; *Lamlom, Irshad & Mosa, 2023*). Excessive salt accumulation in soil leads to the degradation of vital soil properties, including structure, infiltration, porosity, and bulk density (*Du et al., 2019*). This, in turn, affects plant growth and crop productivity through osmotic deregulation, specific ion toxicity, and nutrient imbalances in salt-affected soils (*Khan et al., 2023*). Particularly, extreme osmotic deregulation results in water deficiencies within plants, ultimately reducing yields (*Naveed et al., 2020*). Another major effect is salinity-induced oxidative stress, which stimulates the production of ROS in plant cells (*Ren et al., 2024*; *Zia-ur-Rehman et al., 2022*). Furthermore, relatively high concentrations of sodium (Na) and chloride (Cl) not only cause specific ion toxicity but also disrupt the uptake of essential nutrients such as potassium (K) and calcium (Ca) (*Isayenkov & Maathuis, 2019*). This disruption further exacerbates the negative effects on plant health and growth under saline conditions.

Studies have shown that proline (Pro) and glycine betaine (GB) play essential roles as compatible osmolytes in helping plants cope with various abiotic stressors, including salinity, drought, heavy metals, and low temperatures (*Khatun et al., 2020*; *Bai et al., 2022*). Exogenous applications of Pro and GB, along with seed priming, have proven beneficial in alleviating stress in diverse crops. For example, Pro and GB have been effective in mitigating abiotic stresses in crops like canola (*Iqbal et al., 2022*), cucumber (*Khalid et al., 2022*), rice (*Tania et al., 2022*), and wheat (*Mamedi et al., 2022*). Specifically, *De Freitas et al. (2018)* and *Rady et al. (2019)* reported that exogenous Pro treatments boosted biomass, soluble protein content, and grain yield in wheat and maize. Pro supplementation reduced selenium toxicity in tobacco BY-2 cells by minimizing oxidative damage and enhancing antioxidant enzyme activity (*Khatun et al., 2020*).

GB has similarly demonstrated its potential as an exogenous osmolyte in improving stress tolerance. For instance, GB treatment was shown to increase salt tolerance in maize by modulating morpho-physiological traits, antioxidant defenses, and ionic homeostasis (*Dustgeer et al., 2021*; *Zhu et al., 2022*). GB also protected cotton seedlings from salt stress by enhancing photosynthetic efficiency (*Hamani et al., 2021*). Despite this evidence, no studies, to our knowledge, have explored the combined effects of Pro and GB on reducing

salt stress in crops like maize, suggesting a research gap in understanding the potential synergy of these osmolytes for stress tolerance.

GB is a suitable solute for osmoregulation that can play a crucial role in protecting plants from salt, drought, and excessive temperature stress (*Cha-Um, Samphumphuang & Kirdmanee, 2013*). GB protects plant cells from salt stress through osmotic adjustment (*Gadallah, 1999*), protein stabilization (*Mäkelä, Kärkkäinen & Somersalo, 2000*), preservation of the photosynthetic apparatus (*Allakhverdiev et al., 2003*), and reduction in the amount of oxygen radical scavengers (*Heuer, 2003*). Rice is considered a non-accumulator of GB, though small amounts have been detected in certain rice cultivars under salt stress, including KDML105 (*Chen & Murata, 2008*; *Oh, Chun & Lee, 2003*). As a result, numerous studies have aimed to increase GB biosynthesis in rice to improve salt stress (*Shirasawa, Kishitani & Nishio, 2004*). Prior studies have investigated the exogenous application of GB on rice seedling stage before exposure to salt stress resulted in positive responses (*Rhaman et al., 2024*). The utilization of exogenous GB is not limited to enhancing salt tolerance in rice, as it has also been explored as a means of improving heat tolerance (*Mohammed & Tarpley, 2009*), freezing tolerance (*Chen & Murata, 2008*), and drought tolerance (*Zhu, 2016*).

The exogenous application of proline has been widely recognized as a valuable strategy for increasing plant stress tolerance, primarily because of its osmoprotective properties (*Rasheed et al., 2014*). This phenomenon is observed in various plant species growing under saline conditions, which exhibit improved growth and osmoprotection with exogenously applied proline (*Heuer, 2003*). In rice, the exogenous application of 30 mM proline has been shown to increase early seedling growth under salinity stress. However, excessively high concentrations of proline have been found to impair growth (*Akinmolayan & Adejumo, 2022*). The halophyte species *Allenrolfea occidentalis* exhibited increased growth and reduced ethylene production in response to salt or drought stress due to the exogenous application of proline (*An et al., 2013*). Additionally, proline can increase the activity of antioxidants and protect cell membranes from salt-induced oxidative stress (*Banu et al., 2009*). The exogenous application of 10 mM proline to tobacco suspension cells, under salt stress, resulted in growth promotion through the protection of enzymes and membranes (*Cha-Um, Nhung & Kirdmanee, 2010*). Additionally, exogenous application of proline in soybean cell cultures under salt stress showed an increase in the activity of superoxide dismutase and peroxidase, which are known to play crucial roles in enhancing salt tolerance (*He et al., 2000*). Also, as observed in barley embryo cultures grown under saline conditions, exogenous proline application resulted in reduced $Na^+$ and $Cl^-$ accumulation and increased growth, possibly due to plasma membrane stabilization (*Latef, Hasanuzzaman & Tahjib-Ul-Arif, 2021*).

To address the gap in understanding how combined applications of GB and proline can enhance salt tolerance in rice, this study provides a comprehensive evaluation of these osmolytes' synergistic effects on growth, yield, and biochemical traits under salt stress. While previous research has individually highlighted the role of either GB or proline in mitigating salinity stress across various crops, there is limited insight into their combined impact specifically on rice a crop that is both vital and highly sensitive to salinity. This

investigation is thus unique in its focus on the dual application of GB and proline, offering new perspectives on optimizing osmolyte treatment for improved rice resilience. Our findings contribute novel evidence to the field, potentially informing future strategies for enhancing rice productivity in saline environments.

The current study was designed to assess the effects of exogenous GB and proline on the growth, yield, and chemical attributes of rice plants under salt stress. Our results demonstrate substantial improvements in growth parameters, yield, and biochemical traits, emphasizing the potential of GB and proline applications to enhance rice resilience to salinity stress. Given the increasing challenges associated with climate change and soil salinization, these findings highlight the importance of such approaches in sustaining rice productivity in vulnerable regions, offering a practical solution for improving crop performance under adverse environmental conditions.

## MATERIALS AND METHODS

### Experimental field and soil analysis

The field trials were carried out at the Experimental Station Farm (31°12′20.71″N, 29°55′28.2936″E) of the Faculty of Agriculture, Saba-Basha, Alexandria University, over two consecutive growing seasons of 2021 and 2022. The cultivar Sakha 108 is a high-yielding rice variety developed in Egypt that is adapted to limited water availability and resistant to pests and diseases, while also high-yielding, exhibits sensitivity to salt stress. The experimental site is located in a region with a hot, humid, and arid climate in summer and cool, dry, and windy weather in winter (Fig. 1). The physicochemical properties of the soil were analyzed during both seasons and are presented in Table 1. Soil samples were collected from each plot at two different depths (up to 25 cm) *via* a 2.5 cm spiral auger, and three subsamples were taken from each plot to create a composite sample. The samples were then ground into a fine powder and analyzed for soil organic carbon (as a percentage), available nitrogen, phosphorus, and potassium (in mg/kg) after being oven-dried at 40 °C and passed through a 2 mm filter. The electrical conductivity (EC) was measured *via* standard methods, and the soil pH was determined *via* the technique described by *Carter & Gregorich (2007)*.

### Experimental layout and treatments

In this study, a split-plot design with three replicates was implemented. The glycine betaine treatments were assigned to the main plots, whereas the subplots were subjected to proline treatments. Each subplot covered an area of 10.5 m$^2$, measuring 3.5 m in length and 3 m in width. The application of GB at concentrations of 0 mM control (CK), 10, 20, and 30 mM was carried out *via* foliar spraying throughout three rounds at 30, 45, and 60 days after sowing (DAS). Similarly, the subplots were sprayed with four different concentrations of proline (0 mM (CK), 10, 20, and 30 mM) during the same periods. The spraying application was done at the rates of 300, 450, and 600 L ha$^{-1}$ litres/hectare in the three times of the application, respectively. The GB and proline solutions were prepared by dissolving each compound in distilled water to achieve the desired concentrations, as specified in the experimental design. The spraying of GB and proline was conducted at

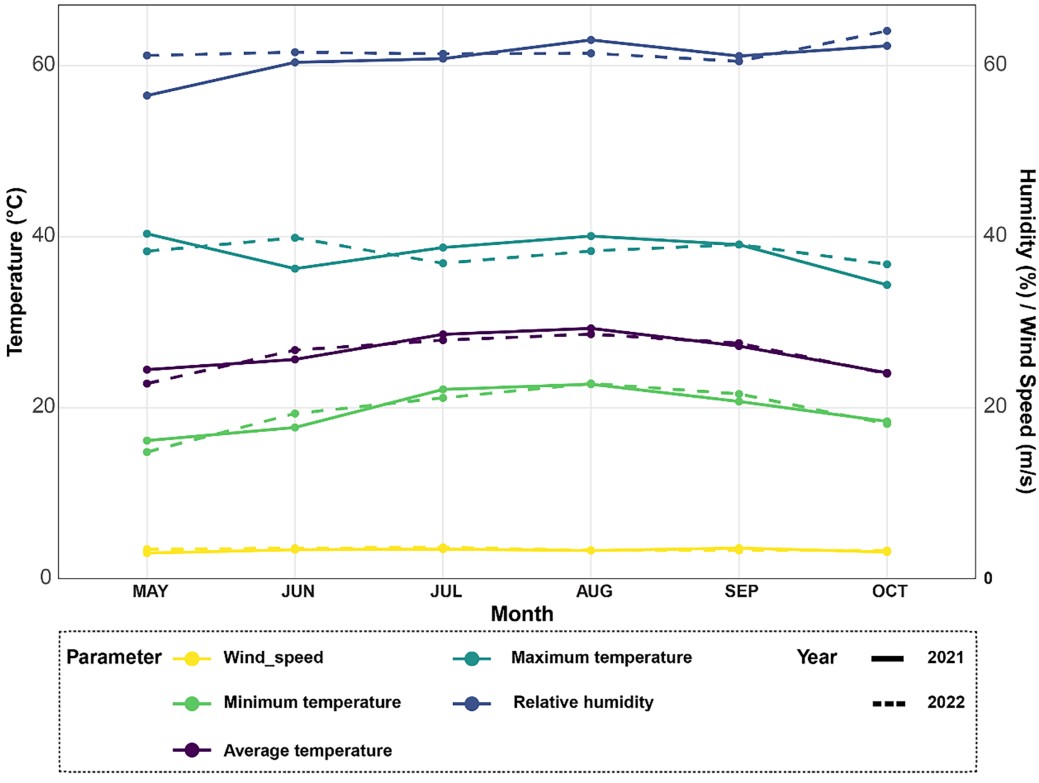

**Figure 1 Climate data analysis revealing variations in average temperature, minimum and maximum temperatures, relative humidity, and wind speed over the 2021–2022 rice seasons for the experimental site.**

early morning hours (typically between 7 a.m. and 9 a.m.) to minimize evaporation and enhance absorption, allowing the plants adequate time to take up the solutions before the midday sun intensified. This timing also helped reduce the risk of phototoxicity and ensured optimal effectiveness of the treatments. The experimental site soil exhibited moderate salinity levels, with EC values of 2.686 and 2.786 dS/m (*Abrol, Yadav & Massoud, 1988*). While these values fall below the conventional threshold for highly saline soils (4 dS/m), they represent conditions that can significantly impact salt-sensitive crops such as rice (*Grattan et al., 2002*).

Seeds were sown directly in the field on May 20, during two consecutive rice-growing seasons, 2021 and 2022 with total duration of 137 days. Each plot consisted of seven rows, with plants spaced 20 cm apart within rows and between rows. The recommended agronomic practices, including fertilization, pest control, and irrigation management, were followed to ensure optimal crop growth throughout the growing period. Nitrogen (N) fertilizer was applied in three doses totaling 180 kg ha$^{-1}$ (from urea with 46.5% N) throughout the plant growth stages. Before planting, the field received a base dose of 150 kg ha$^{-1}$ $P_2O_5$ and 100 kg ha$^{-1}$ $K_2O$.

**Table 1 Initial soil properties of the experimental soil during the 2021 and 2022 seasons.**

| Parameters | | Values | |
| --- | --- | --- | --- |
| | | 2021 season | 2022 season |
| Particle size distribution | Clay % | 17.5 | 19.6 |
| | Sand % | 70 | 68.5 |
| | Silt % | 12.5 | 11.9 |
| | Texture grade | Silt loam | Silt loam |
| Available nutrients | N (mg.kg$^{-1}$) | 77.48 | 89.28 |
| | P (mg.kg$^{-1}$) | 30.35 | 35.84 |
| | K (mg.kg$^{-1}$) | 379.32 | 436.53 |
| Soluble cations and anions (cmol.kg$^{-1}$ soil) | Ca$^{++}$ | 2.08 | 1.94 |
| | Mg$^{++}$ | 4.69 | 4.75 |
| | Na$^{+}$ | 18.12 | 18.50 |
| | K$^{+}$ | 0.72 | 0.77 |
| | HCO$_3^-$ | 3.67 | 3.73 |
| | Cl$^-$ | 15.53 | 15.66 |
| | SO$_4^-$ | 6.41 | 6.57 |
| Chemical properties | pH (Susp. 1:2.5 soil–water) | 7.23 | 7.26 |
| | EC (dSm$^{-1}$) | 2.686 | 2.786 |
| | C.E.C (cmol.kg$^{-1}$) | 18.76 | 19.38 |
| | Organic matter (%) | 1.38 | 1.73 |
| | CaCO$_3$ (%) | 43.22 | 42.32 |

## Meteorological parameters

The meteorological station in Egypt's Alexandria Governorate provided the data for the summers of 2021 and 2022 (Fig. 1). There was little rain and arid conditions in the experimental field. The average high air temperature from May to September was 36.4 °C, while the average low air temperature from the same growth season was 22.35 °C. Between 35 and 42 percent relative humidity was experienced throughout this time.

## Data collection

### Growth, yield, and yield characteristics

Plant height (cm) was assessed by measuring the distance from the soil surface to the top of the ten randomly selected plants at harvest. The dry weight was determined by measuring the dry weight of ten random plants from each plot, and the average total weight was recorded. The tiller number, number of spikes per m$^2$, spike length (cm), and number of spikelets per spike were determined from ten plants taken from each sub-plot. The 1,000-kernel weight (g) was obtained from the weight of 1,000 grains for three samples for each sub-plot, then the average was subsequently calculated. After harvesting, the biological yield (t/ha) was quantified by weighing all harvested plants (straw and grains) and subsequently converted to obtain the value into t/ha for each sub-plot. The grain yield

(t/ha) was also determined for all the plants in each sub-plot and then converted to the grain yield (t/ha). The harvest index (%) was calculated according to the following formula:

$$\text{Harvest index (\%)} = \frac{\text{Grain yield}}{\text{Biological yield}} \times 100$$

### Chemical analysis

During harvest, chemical analysis of the rice grains was performed to determine their crude grain protein content and amylose content. The crude protein content was calculated using the formula: Crude protein % = Nitrogen (N) × 6.25. The N content of the rice grains was determined *via* the micro-Kjeldahl method described by *Mariotti, Tomé & Mirand (2008)*. To determine the amylose content, a sample of 100 mg was accurately weighed into a 100 ml volumetric flask, and 1 ml of 95% ethanol and 9 ml of 1 N NaOH were added. The mixture was then heated for 10 min in a boiling water bath to gelatinize the starch, cooled, and brought to volume with water (*Williams et al., 1958*). A 5 ml portion of the gelatinized starch solution, 1 ml of 1 N acetic acid, and 2 ml of iodine solution were added, and the volume was adjusted to 100 ml with distilled water. The mixture was then shaken and allowed to stand for 20 min before the transmission was read at 620 nm *via* a UV-visible spectrophotometer (Camspec model-M330B; Camspec, Crawley, England).

## Statistical analysis

The data for all measured traits were analyzed through analysis of variance (ANOVA) using SAS 9.4 software (SAS Institute Inc., Cary, NC, USA) following the methodology outlined by *Gomez & Gomez (1984)*. Treatment means were compared using the least significant difference (LSD) test at $p < 0.05$ and $p < 0.01$. All figures were generated with RStudio 4.1.1 (Integrated Development for RStudio, Inc., Boston, MA, USA, www.rstudio.com). The weather data across the two growing seasons were visualized using the ggplot2 package (*Wickham, 2011*). Principal component analysis (PCA) was performed with the FactoExtra package (*Kassambara & Mundt, 2016*), while heatmaps were generated using the ComplexHeatmap package (*Gu, Eils & Schlesner, 2016*) for enhanced visualization. Pearson correlation analysis was carried out and visualized with the ggplot2, reshape2 (*Wickham, 2020*), and RColorBrewer (*Neurwirth, 2014*) packages. Path analysis was conducted using the lavaan package (*Rosseel, 2012*) to specify and fit the structural equation model, with visualization of the path diagram done using the semPlot package (*Epskamp, 2015*).

## RESULTS

### Climatic conditions during the rice growing seasons of 2021 and 2022

Climate data was collected from May to October 2021 and 2022 to analyze growing conditions for rice plants (Fig. 1). In 2021, average monthly temperatures ranged from 22 °C in May up to a peak of 32 °C in August. Temperatures were slightly higher in 2022, ranging from 24 °C in May with maximum monthly averages reaching 34 °C in both July

and August. For the relative humidity levels, it also varied between the two years, as it averaged 65% in June and increased to a high of 78% in October in 2021. Similarly in 2022, humidity levels averaged between 65–80%, however peaked at the higher value of 82% in August. In terms of temperature extremes, maximum monthly averages ranged from 30 °C in May 2021 up to 36 °C in both July and August 2022. Meanwhile, minimum monthly temperatures stayed above 18 °C for the duration of both growing seasons. Fluctuations in temperature and humidity between the months can impact rice physiological processes like photosynthesis and grain-filling. Wind speeds during the growing season followed a comparable trend in both years, as moderate speeds of 10–15 kmph were typical from May through September.

## Impact of GB and PRO on rice agronomic performance

The application of exogenous GB and Pro, as well as their interaction, significantly influenced various agronomic, yield, and chemical attributes, including dry weight (DW), plant height (PH), tiller number (NT), number of spike per meter (NSM), spike length (SL), number of spikelets per spike (NSS), 1,000-kernel weight (TKW), biological yield (BY, ton/ha), grain yield (GY, ton/ha), harvest index (HI%), amylose percentage (AC%), and protein content (PC%) under salt stress during both the 2 years of study (Tables 2 and 3).

The 30 GB treatment resulted in the highest dry weight, with 162.2 g in 2021 and 169.7 g in 2022, compared to the control (CK) treatment, which had 106.67 g in 2021 and 122.83 g in 2022. Similarly, the 30 Pro treatment showed the highest dry weight, with 162.16 g in 2021 and 166.01 g in 2022, compared to the CK treatment (120.67 g in 2021 and 122.33 g in 2022). Plant height was significantly increased in the 30 GB and 30 Pro treatments, with 88.94 cm and 99.00 cm in 2021, and 88.94 cm and 99.00 cm in 2022, respectively, compared to CK treatment (67.38 cm in 2021 and 72.18 cm in 2022). The number of tillers also increased significantly, with the 30 GB treatment showing 10.58 tillers in 2021 and 10.33 tillers in 2022, while the 30 Pro treatment showed 9.94 tillers in 2021 and 11.00 tillers in 2022, compared to CK treatment (6.31 tillers in 2021 and 6.50 tillers in 2022). The number of spikes per square meter was highest for 30 GB and 30 Pro treatments, with 212.22 spikes/m$^2$ in 2021 and 212.32 spikes/m$^2$ in 2022 for GB, and 220.63 spikes/m$^2$ in 2021 and 218.62 spikes/m$^2$ in 2022 for Pro, while the CK treatment had 145.71 spikes/m$^2$ in 2021 and 144.60 spikes/m$^2$ in 2022. Spike length was significantly higher under the 30 GB and 30 Pro treatments, with 22.05 cm and 21.33 cm in 2021, respectively, compared to CK treatment (11.01 cm in 2021 and 13.13 cm in 2022). The number of spikelets per spike also increased significantly under higher GB and Pro treatments, with 13.52 spikelets in the 30 GB treatment and 12.91 spikelets in the 30 Pro treatment in 2021, and 10.33 spikelets in the 30 GB treatment and 9.80 spikelets in the 30 Pro treatment in 2022, compared to CK treatment (6.75 spikelets in 2021 and 8.87 spikelets in 2022). The interaction between GB and Pro was significant for all traits, demonstrating a synergistic effect in improving rice performance under saline conditions (Table 2). For 1,000-kernel weight, the highest values were observed in the 30 GB treatment, with 49.58 g in 2021 and 50.87 g in 2022, compared to the CK which showed 26.66 g in 2021 and 42.41 g in 2022. Biological yield was also

**Table 2 Mean effects of glycine betaine (GB) and proline (Pro) on the growth and agronomic attributes of rice (*Oryza sativa* L.) plants during two growing seasons.**

| | Dry weight (g) | | Plant height (cm) | | Tiller number | | No. spikes/m$^2$ | | Spike length (cm) | | No. spikelet/spike | |
|---|---|---|---|---|---|---|---|---|---|---|---|---|
| | 2021 | 2022 | 2021 | 2022 | 2021 | 2022 | 2021 | 2022 | 2021 | 2022 | 2021 | 2022 |
| CK | 106.67d | 122.83d | 67.38c | 72.18d | 6.31d | 6.50c | 145.71d | 144.60d | 11.01d | 13.13b | 6.75d | 8.87d |
| 10 GB | 137.08c | 138.69c | 79.47b | 71.69c | 7.11c | 8.15b | 175.94c | 175.60c | 16.30c | 15.29a | 10.38c | 10.11a |
| 20 GB | 157.55b | 158.83b | 81.52b | 83.65b | 8.72b | 8.38b | 198.55b | 197.22b | 18.94b | 16.08a | 12.77b | 10.41a |
| 30 GB | 162.2a | 169.7a | 88.94a | 99.00a | 10.58a | 10.33a | 212.22a | 212.32a | 22.05a | 16.00a | 13.52a | 10.33a |
| LSD | 1.42 | 2.65 | 2.52 | 2.42 | 0.85 | 0.91 | 2.42 | 2.79 | 0.80 | 1.48 | 0.63 | 0.76 |
| Mean | 140.87 | 147.51 | 79.32 | 81.63 | 8.18 | 8.34 | 183.10 | 182.43 | 17.07 | 15.12 | 10.85 | 9.93 |
| CK | 120.67d | 122.33d | 70.05c | 72.18c | 6.88b | 7.36b | 160.61c | 159.80c | 14.11c | 13.43c | 9.05c | 8.88c |
| 10 Pro | 132.86c | 142c | 66.25d | 82.52b | 6.77b | 7.05b | 166.05c | 195.21d | 14.86c | 17.24a | 10.97b | 11.19a |
| 20 Pro | 147.83b | 159.72b | 81.97b | 81.33b | 9.10a | 7.94b | 195.14b | 196.80b | 18.00b | 15.78b | 10.50b | 9.85b |
| 30 Pro | 162.16a | 166.01a | 99.05a | 90.74a | 9.94a | 11.00a | 220.63a | 218.62a | 21.33a | 14.01c | 12.91a | 9.80b |
| LSD | 3.19 | 4.54 | 2.79 | 2.62 | 0.88 | 0.89 | 2.14 | 1.79 | 1.36 | 1.31 | 0.97 | 0.78 |
| Mean | 140.88 | 147.51 | 79.33 | 81.69 | 8.17 | 8.34 | 185.60 | 192.60 | 17.07 | 15.11 | 10.86 | 9.93 |
| GB | ** | ** | ** | ** | ** | ** | ** | ** | ** | ** | ** | ** |
| Pro | ** | ** | ** | ** | ** | ** | ** | ** | ** | ** | ** | ** |
| GB*Pro | ** | ** | ** | ** | ** | ** | ** | ** | ** | ** | ** | ** |

**Note:**
CK, control; GB, glycine betaine; Pro, proline. Different small letters in each column indicate statistically significant differences. Two asterisks (**) indicates significance at 0.1% (*p* 0.01) probability level, respectively.

highest under the 30 GB treatment, with 10.52 t/ha in 2021 and 10.43 t/ha in 2022, while the control had 7.49 t/ha in 2021 and 7.04 t/ha in 2022. For grain yield, the 30 GB treatment recorded the highest values of 4.22 t/ha in 2021 and 4.30 t/ha in 2022, compared to the control with 2.57 t/ha in 2021 and 2.64 t/ha in 2022. The harvest index also increased under the GB treatments, with the 30 GB treatment showing 39.58% in 2021 and 40.36% in 2022, compared to the control with 34.28% in 2021 and 37.07% in 2022. Amylose content was highest in the 30 GB treatment, with 21.44% in 2021 and 19.34% in 2022, while the control showed 13.01% in 2021 and 14.46% in 2022. Protein content was also enhanced by GB treatment, with the 30 GB treatment showing 10.78% in 2021 and 9.66% in 2022, compared to the control with 7.57% in 2021 and 7.53% in 2022 (Table 3). For Pro, the highest 1,000-kernel weight was recorded in the 30 Pro treatment, with 53.25 g in 2021 and 48.91 g in 2022, compared to the control with 32.0 g in 2021 and 44.27 g in 2022. Biological yield also increased with Pro application, with the 30 Pro treatment showing 11.39 t/ha in 2021 and 11.73 t/ha in 2022, compared to the control with 8.31 t/ha in 2021 and 7.90 t/ha in 2022. Grain yield was highest under the 30 Pro treatment, with 5.21 t/ha in 2021 and 5.52 t/ha in 2022, compared to the control with 2.91 t/ha in 2021 and 3.07 t/ha in 2022. The harvest index was significantly higher in the 30 Pro treatment, with 45.44% in 2021 and 46.85% in 2022, compared to the control with 34.83% in 2021 and 38.60% in 2022. Amylose content was highest in the 30 Pro treatment, with 20.03% in 2021 and 18.08% in 2022, while the control showed 12.34% in 2021 and 16.34% in 2022. Protein content also

**Table 3 Mean effects of glycine betaine (GB) and proline (Pro) on the chemical and yield attributes of rice (*Oryza sativa L.*) plants during two growing seasons.**

| | 1,000-Kernel weight (g) | | Biological yield (t/ha) | | Grain yield (t/ha) | | Harvest index (%) | | Amylose content (%) | | Protein content (%) | |
|---|---|---|---|---|---|---|---|---|---|---|---|---|
| | 2021 | 2022 | 2021 | 2022 | 2021 | 2022 | 2021 | 2022 | 2021 | 2022 | 2021 | 2022 |
| CK | 26.66d | 42.41c | 7.49c | 7.04c | 2.57c | 2.64c | 34.28b | 37.07b | 13.01d | 14.46c | 7.57d | 7.53c |
| 10 GB | 37.76c | 46.70b | 9.45b | 9.13b | 3.71b | 3.87b | 38.89a | 41.15a | 17.64c | 17.99b | 8.78c | 8.17b |
| 20 GB | 42.75b | 47.67b | 9.56b | 9.96a | 4.08a | 4.13a | 41.83a | 40.96a | 19.11b | 19.67a | 9.91b | 9.23ab |
| 30 GB | 49.58a | 50.87a | 10.52a | 10.43a | 4.22a | 4.30a | 39.58a | 40.36ab | 21.44a | 19.34a | 10.78a | 9.66a |
| LSD | 2.88 | 2.18 | 0.47 | 0.55 | 0.28 | 0.23 | 3.51 | 3.34 | 0.83 | 0.73 | 0.52 | 0.53 |
| Mean | 39.18 | 46.91 | 9.25 | 9.14 | 3.64 | 3.73 | 38.64 | 39.88 | 17.8 | 17.86 | 9.26 | 8.64 |
| Ck | 32.0c | 44.27b | 8.31c | 7.90c | 2.91c | 3.07c | 34.83c | 38.60b | 12.34c | 16.34c | 8.37c | 8.02c |
| 10 PRO | 30.93c | 46.75ab | 8.33c | 8.33bc | 2.90c | 2.96c | 34.74c | 35.55c | 19.36b | 19.36a | 8.49c | 7.61c |
| 20 PRO | 40.58b | 47.71a | 9.01b | 8.77b | 3.57b | 3.39b | 39.56b | 38.54b | 19.86b | 17.86b | 9.84b | 9.244b |
| 30 PRO | 53.25a | 48.91a | 11.39a | 11.73a | 5.21a | 5.52a | 45.44a | 46.85a | 20.03a | 18.08b | 10.35a | 10.26a |
| LSD | 3.34 | 2.84 | 0.35 | 0.544 | 0.30 | 0.18 | 3.76 | 2.40 | 0.77 | 0.63 | 0.78 | 0.71 |
| Mean | 39.19 | 46.91 | 9.26 | 9.18 | 3.65 | 3.73 | 38.64 | 39.88 | 17.89 | 17.91 | 9.26 | 8.78 |
| GB | ** | ** | ** | ** | ** | ** | ** | ** | ** | ** | ** | ** |
| PRO | ** | ** | ** | ** | ** | ** | ** | ** | ** | ** | ** | ** |
| GB*PRO | ** | ** | ** | ** | ** | ** | ** | ** | ** | ** | ** | ** |

**Note:**
CK, control; GB, glycine betaine; Pro, proline. Different small letters in each column indicate statistically significant differences. Two asterisks (**) indicates significance at 0.1% ($p$ 0.01) probability level, respectively.

increased in the 30 Pro treatment, with 10.35% in 2021 and 10.26% in 2022, compared to the control with 8.37% in 2021 and 8.02% in 2022 (Table 3).

The interaction between GB and Pro had significant effects on the growth and agronomic attributes of rice (*Oryza sativa* L.) during the two growing seasons (Table 4). In 2021, the highest dry weight (184.11 g) was recorded with 30 GB and 30 Pro, followed by 30 GB with 186.56 g in 2022. Conversely, the lowest dry weight values were observed in the control group (0 GB, 0 Pro), with 106.67 g in 2021 and 105.00 g in 2022. Similarly, plant height was significantly affected, with the tallest plants (186.00 cm in 2022) observed under 30 GB and 30 Pro, while the shortest plants (56.55 cm in 2021 and 56.67 cm in 2022) were recorded under the control treatment. For tiller number, the interaction of 30 GB and 30 Pro produced the highest value (15.33 in 2021 and 16.28 in 2022), while the control had the lowest tiller numbers (6.22 in 2021 and 4.67 in 2022). In terms of spike number per square meter, the highest number (264/m$^2$) was achieved with 30 GB and 30 Pro in both years, while the lowest value (135.66/m$^2$ in 2021) was recorded in the control. Spike length was also significantly affected by the treatments. The longest spikes (17 cm in 2021 and 11.77 cm in 2022) were found in the combination of 30 GB and 30 Pro, whereas the shortest spikes (6.33 cm in 2021 and 6.53 cm in 2022) were recorded in the control. For spikelet number, the highest number (28.66 in 2021 and 23.66 in 2022) was found under 30 GB and 30 Pro, and the lowest (11.00 in 2021 and 11.33 in 2022) was observed in the control group. These results highlight the positive impact of the combined application of GB and Pro on improving rice growth and agronomic performance, with the 30 GB and 30 Pro treatment

showing the most significant increases across all measured traits compared to the control and other treatments. For 1,000-kernel weight, the highest value (70.00 g in 2021 and 73.20 g in 2022) was recorded with the combination of 30 GB and 30 Pro, while the lowest value (25.34 g in 2021 and 25.30 g in 2022) was found in the control (0 GB, 0 Pro) treatment. Biological yield showed a similar pattern, with the highest values of 13.59 t/ha in 2021 and 13.30 t/ha in 2022 achieved under 30 GB and 30 Pro, and the lowest (5.41 t/ha in 2021 and 5.41 t/ha in 2022) observed in the control group. Grain yield also significantly improved with higher GB and Pro levels, with the highest grain yield (6.64 t/ha in 2022) under the combination of 30 GB and 30 Pro, compared to the lowest yield (1.85 t/ha in 2021 and 1.85 t/ha in 2022) in the control. The harvest index was highest for the combination of 30 GB and 30 Pro (45.27% in 2021 and 48.94% in 2022), indicating better allocation of biomass towards grain production. The control treatment had the lowest harvest index (32.95% in 2021 and 34.87% in 2022). Amylose content was generally higher in treatments with higher Pro and GB concentrations, with the highest values (24.24% in 2021 and 20.09% in 2022) recorded under 30 GB and 30 Pro. The control group showed lower amylose content (13.00% in 2021 and 12.00% in 2022). Protein content followed a similar trend, with the highest protein content (12.33% in 2021 and 12.00% in 2022) under the 30 GB and 30 Pro treatment, and the lowest (6.43% in 2021 and 6.69% in 2022) in the control. These results highlight the positive influence of GB and Pro treatments on the chemical and yield attributes of rice, with the highest values typically observed with the highest concentrations of both GB and Pro (Table 5).

## Correlation analysis of studied traits

Pearson correlation analysis was conducted to investigate the relationships between various agronomic traits. The results revealed significant positive correlations among several traits (Fig. 2). PH exhibited a strong positive correlation with GY (r = 0.94***), BY (r = 0.92***), HI (r = 0.90***), NT (r = 0.83***), PC (r = 0.90***), and TKW (r = 0.83***). PC was strongly correlated with DW (r = 0.91***), NT (r = 0.90***), NSM (r = 0.89***), GY (r = 0.89***), and BY (r = 0.88***). AC demonstrated significant correlations with TKW (r = 0.81***), SL (r = 0.98***), NSS (r = 0.94***), and HI (r = 0.50*). TKW was highly correlated with DW (r = 0.89***), PH (r = 0.83***), NT (r = 0.90***), BY (r = 0.94***), GY (r = 0.90***), and HI (r = 0.79***). SL showed significant correlations with traits such as DW (r = 0.90***), NSS (r = 0.91***), and BY (r = 0.74**). NSS was positively correlated with DW (r = 0.83***), PH (r = 0.52*), NT (r = 0.69**), and GY (r = 0.66**). NSM displayed significant positive correlations with DW (r = 0.88***), PH (r = 0.83***), and TKW (r = 0.83***). The correlation between HI and GY was strong (r = 0.95***), while HI correlated well with BY (r = 0.86***). GY exhibited robust correlations with a wide range of traits, including DW (r = 0.82**), PH (r = 0.94***), NT (r = 0.84***), and BY (r = 0.97***). BY was positively correlated with DW (r = 0.84***), PH (r = 0.92***), NT (r = 0.87***), and NSM (r = 0.85***).

**Table 4 Interaction effects of glycine betaine (GB) and proline (Pro) on the growth and agronomic attributes of rice (*Oryza sativa* L.) plants during the two growing seasons.**

| GB | Pro | Dry weight (g) | | Plant height (cm) | | Tiller number | | No. spikes/m$^2$ | | Spike length (cm) | | No. spikelet/spike | |
|---|---|---|---|---|---|---|---|---|---|---|---|---|---|
| | | 2021 | 2022 | 2021 | 2022 | 2021 | 2022 | 2021 | 2022 | 2021 | 2022 | 2021 | 2022 |
| 0 | 0 | 106.67j | 105.00i | 56.55f | 56.67d | 6.22e | 4.67d | 138.21h | 135.66h | 6.33e | 6.53c | 11i | 11.33c |
| | 10 | 106.33j | 116.00h | 83.22c | 88.67c | 6.22e | 7.67bcd | 148.44gf | 147.77g | 6.67e | 11.77a | 11.55hi | 18b |
| | 20 | 107.00j | 134.67fg | 68.22de | 68.16d | 7.77cde | 7.77bcd | 147.99g | 147.77g | 7.33e | 8.55bc | 10.44i | 13abc |
| | 30 | 106.67j | 135.67fg | 72.22de | 75.22d | 7.33de | 9.33bc | 148.21g | 147.77g | 6.66e | 8.66 | 11i | 10.20c |
| 10 | 0 | 112.33i | 114.66hi | 58.44f | 70.56d | 6.11e | 6.33cd | 148.21g | 147.21g | 6.67e | 8.67bc | 11i | 10.20c |
| | 10 | 126.55h | 134.44fg | 64.44ef | 70.56d | 6.22e | 6.30cd | 157.21f | 158.19f | 10.89d | 10.57ab | 14.33gh | 18b |
| | 20 | 145.0f | 143.44ef | 68.22de | 68.16d | 7.77cde | 7.77bcd | 183.35d | 185.33d | 11d | 9.44ab | 18.55def | 16.33b |
| | 30 | 164.44d | 162.22d | 73.89d | 77.50d | 7.00de | 7.78bcd | 214.99b | 211.66c | 13bcd | 10.55ab | 21.33cd | 16.63b |
| 20 | 0 | 127.66h | 130.67g | 71.33de | 71.67d | 6.66de | 7.33cd | 172.66e | 171.00e | 12.22cd | 10.89ab | 15.88gf | 15.88bc |
| | 10 | 143.77f | 150.33e | 72.22de | 75.22d | 7.33de | 9.33bc | 172.66e | 171.00e | 12.22cd | 10.98ab | 15.88fg | 15.88bc |
| | 20 | 165.33d | 174.77bc | 86.67c | 89.67c | 9.77c | 7.16cd | 193.55c | 195.33d | 11.67d | 10.44ab | 19.66de | 16.89b |
| | 30 | 193.44a | 179.56ab | 97.67b | 98bc | 12.67b | 7.94bcd | 255.33a | 251.55b | 15ab | 9.44ab | 24.33b | 15.66bc |
| 30 | 0 | 136.00g | 139.00fg | 83.22c | 88.67c | 6.22e | 7.66bcd | 183.35d | 185.33d | 11d | 9.44ab | 18.55def | 16.33b |
| | 10 | 154.78e | 167.44cd | 98b | 95.67bc | 8.67cd | 9.27bc | 145.88gh | 143.88gh | 14.11bc | 10.33ab | 17.66ef | 17.11b |
| | 20 | 174.01c | 186.00a | 103ab | 99.33b | 9.56c | 10.78b | 255.66a | 255.98ab | 12cd | 11ab | 23.33bc | 18.89b |
| | 30 | 184.11b | 186.56a | 112a | 112.33a | 15.33a | 16.28a | 264a | 264.05a | 17a | 11.77a | 28.66a | 23.66a |

**Note:**
GB, glycine betaine; Pro, proline. Different small letters in each column indicate statistically significant differences.

## Path analysis of agronomic and chemical traits on grain yield

The path analysis results for the direct effects of agronomic traits GY demonstrated that several traits had statistically significant impacts (Fig. 3). TKW exhibited the highest significant positive direct effect on GY (0.43). Similarly, BY showed a significant positive direct effect on GY (0.42), highlighting the importance of biomass in determining yield potential. Furthermore, HI significantly and positively influenced GY (0.31), emphasizing the role of the harvest index in yield optimization. Also, NSS had a significant positive direct effect on GY (0.25). On the other hand, SL had a significant negative direct effect on GY (−0.30), suggesting that an increase in the number of spikelets may negatively impact grain yield. Also, the other traits, including AC, PC, NSM, NT, PH, and DW had non-significant negative direct effects on GY, indicating that their influence on yield is relatively minor and not statistically meaningful.

## Synergistic effects of glycine betaine and proline on agronomic and biochemical traits of rice

The heatmap clustering analysis revealed distinct patterns in the response of rice traits to various combinations of GB and Pro treatments (Fig. 4). The traits clustered into three major groups based on their responses to the treatments. The first cluster included NSM, DW, and PH. These traits showed a consistent response pattern, with the combination of 30P_30GB leading to the most significant enhancement, particularly for PH and NSM,

**Table 5 Interaction effects of glycine betaine (GB) and proline (Pro) on the chemical and yield attributes of rice (*Oryza sativa L.*) during two growing seasons.**

| GB | Pro | 1,000-Kernel weight (g) | | Biological yield (t/ha) | | Grain yield (t/ha) | | Harvest index (%) | | Amylose content (%) | | Protein content (%) | |
|---|---|---|---|---|---|---|---|---|---|---|---|---|---|
| | | 2021 | 2022 | 2021 | 2022 | 2021 | 2022 | 2021 | 2022 | 2021 | 2022 | 2021 | 2022 |
| 0 | 0 | 26.66gh | 25.34gh | 5.74f | 5.41g | 1.89f | 1.85g | 32.95cd | 34.87cd | 13.00ef | 12.00b | 6.69f | 6.43g |
| | 10 | 28gh | 23.2h | 7.76ef | 7.22egf | 2.32ef | 2.33fg | 36.49bcd | 32.40d | 12.33f | 18.89a | 6.99f | 7.02fg |
| | 20 | 25.33h | 24.00h | 7.12ef | 6.66gf | 2.67def | 2.45fg | 37.78bcd | 36.84abcd | 13.66ef | 13.66b | 8.22def | 8 defg |
| | 30 | 26.66gh | 25.30h | 9.34cd | 8.88cde | 3.40cd | 3.92cd | 36.49bcd | 44.18abcd | 13.00ef | 14.00a | 8.40def | 8.68cdef |
| 10 | 0 | 26.66gh | 26.76gh | 7.76de | 8.88cde | 3.40cd | 3.92cd | 29.92d | 44.18abcd | 13.00ef | 14.00 | 8.40def | 8.68cdef |
| | 10 | 27.10gh | 26.10gh | 7.81de | 8.07def | 2.91def | 2.81ef | 37.27bcd | 35.53cd | 18.89cd | 18.89a | 7.66ef | 7.69efg |
| | 20 | 42.33def | 44.23def | 8.69cde | 8.54cdef | 3.03de | 3.13def | 35.01bcd | 36.91abcd | 19.30c | 19.30a | 9.40bcde | 9.09bcde |
| | 30 | 54.66bc | 56.37bc | 9.06cd | 11.75ab | 5.24ab | 5.63b | 46.78ab | 47.96ab | 19.38c | 19.79a | 9.66bcd | 9.37bcde |
| 20 | 0 | 32.33fgh | 35.22fgh | 8.69cde | 8.79cdef | 3.03de | 3.38cde | 34.87bcd | 38.43abcd | 16.00de | 19.55a | 9.00cde | 7.88defg |
| | 10 | 32.33fgh | 34.33fgh | 9.46cd | 8.79cde | 3.31cde | 3.38cde | 34.87bcd | 38.43abcd | 16.00de | 20.09a | 9.00cde | 7.88defg |
| | 20 | 44.66cde | 44.28def | 9.19cd | 9.54cd | 4.09c | 3.88cd | 44.36abc | 40.66abcd | 20.77bc | 19.50a | 10.66abc | 10.18abc |
| | 30 | 61.66ab | 61.67ab | 11.70ab | 12.70a | 6.17a | 5.89ab | 53.22a | 46.33abc | 23.66ab | 18.99a | 11ab | 11ab |
| 30 | 0 | 42.33def | 44.02def | 9.46cd | 8.54cdef | 3.31cde | 3.13def | 35.01bcd | 36.91abcd | 19.30c | 19.3a | 9.40bcde | 9.09bcde |
| | 10 | 36.00efg | 38.22ef | 11.21b | 9.23cde | 3.34cde | 3.31cde | 36.92bcd | 35.83bcd | 19.56c | 19.56a | 10.33bc | 7.87defg |
| | 20 | 50.00cd | 62.2a | 10.25bc | 10.36cd | 4.20bc | 4.11c | 41.12abcd | 39.75abcd | 22.66ab | 18.98a | 11.07ab | 9.71abc |
| | 30 | 70.00a | 73.2a | 13.3a | 13.59a | 6.03a | 6.64a | 45.27abc | 48.94a | 24.24a | 20.09a | 12.33a | 12a |

**Note:**
CK, control; GB, glycine betaine; Pro, proline. Different small letters in each column indicate statistically significant differences.

indicating a strong synergistic effect of this treatment. The second cluster comprised BY, PC, NT, NSS, AC, SL, and GY. These traits exhibited a similar response across the different treatment combinations. The treatment combination of 30P_30GB had a particularly notable impact on NT and GY, suggesting that moderate levels of GB combined with high Pro concentrations are optimal for improving grain yield and its related components. The third cluster comprised HI and TKW. These traits showed a distinct response, with the 30P_30GB combination enhancing both HI and TKW significantly more than other treatments, highlighting the importance of high GB and Pro levels for optimizing these traits. The hierarchical clustering of treatments emphasized that combinations with higher Pro levels (such as 20P_30GB, 30P_20GB, and 30P_30GB) tend to group together and have more pronounced positive effects on most traits compared to treatments with lower Pro concentrations or GB alone. CK and those with only one component (*e.g.*, 10GB, 20P) generally displayed weaker responses, underscoring the benefit of combined GB and Pro application for enhancing rice traits.

## Principal component analysis of the combined effects of glycine betaine and proline on rice traits

The PCA performed on the combined treatments of GB and Pro applied to rice reveals distinct clustering patterns, indicating the influence of these treatments on the 12 studied traits (Fig. 5). The first principal component (Dim1) captures 82.1% of the total variation,
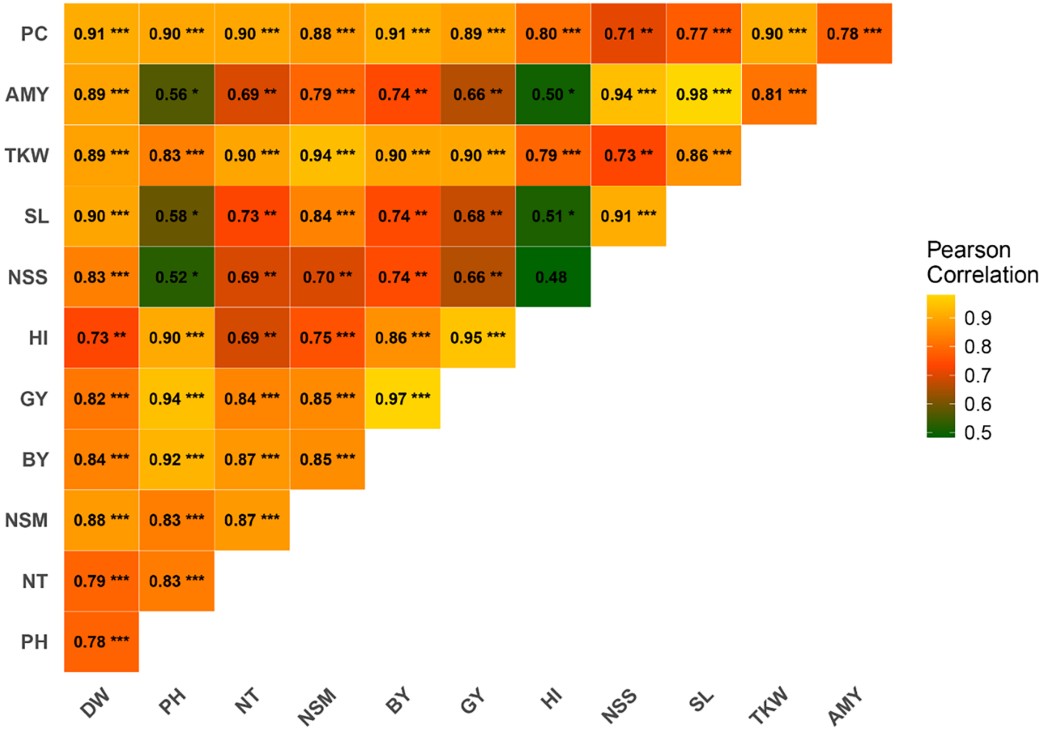

**Figure 2 Correlation among rice traits under the application of foliar spraying of glycine betaine and proline under saline conditions.** The blue and red colors indicate positive and negative correlations, respectively. A higher or lower correlation among parameters is linked to color intensity. DW, dry weight; PH, plant height; NT, tiller number; NSM, number of spikes m-2; SL, spike length; NSS, number of spikelets per spike; TKW, 1,000-kernel weight; GY, grain yield; BY, biological yield; HI, harvest index; AC, amylose content; PC, protein content. *, **, *** indicate significant at 5% ($p \leq 0.05$), significant at 0.1% ($p \leq 0.01$) and significant at 0.01% ($p \leq 0.001$) probability level, respectively.

while the second component (Dim2) accounts for 10.3%. The biplot shows that treatments such as 30P_10GB, 30P_20GB, and 30P_30GB cluster together, indicating similar effects on the traits, particularly on NT, PC, BY, and GY, which are positively associated with Dim1. Conversely, treatments such as 20P_20GB and 20P_30GB are more closely related to traits like DW, TKW, and NSM, which align moderately with Dim1. CK, 10P, 30P, and 10GB are distinctly separated from the treated combinations, especially along Dim1. Traits such as PH and HI show negative associations with Dim1 across most treatments, suggesting a contrasting response compared to other traits like NSS, AC, and SL.

## DISCUSSION

The application of exogenous GB and Pro under salt stress resulted in significant enhancements in various agronomic, yield, and chemical traits of rice, suggesting that these osmoprotectants play a crucial role in mitigating salt-induced damage. For instance, the observed increases in DW, PH, NT, and spikelet traits across both seasons suggest that GB and Pro enhance the plant's ability to maintain cellular turgor, stabilize proteins, and reduce oxidative stress, which are critical responses to salt stress (*Jain et al., 2021*). The improved biomass accumulation, as evidenced by higher DW, is likely due to the osmotic

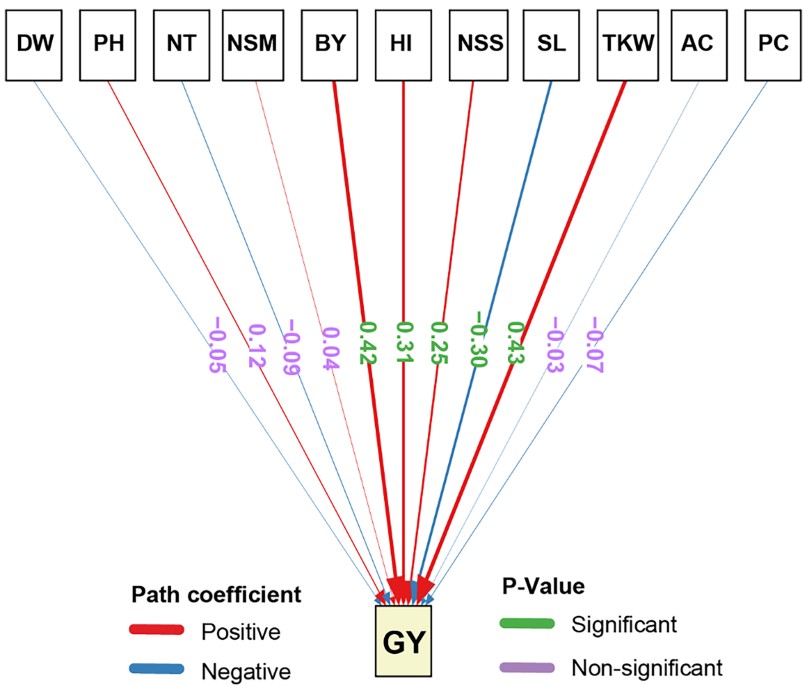

**Figure 3 Path analysis of the direct and indirect effects on the studied traits of rice under salt stress.** Positive coefficients are indicated in blue and negative coefficients in red. *P*-values are shown in green for significant and purple for non-significant paths. The width of the arrows represents the magnitude of the path coefficients, providing a visual representation of the strength of each relationship. DW, dry weight; PH, plant height; NT, tiller number; NSM, number of spikes m-2; SL, spike length; NSS, number of spikelets per spike; TKW, 1,000-kernel weight; GY, grain yield; BY, biological yield; HI, harvest index; AC, amylose content; PC, protein content.

adjustment facilitated by GB and Pro, which improves water uptake and retention under saline conditions (*Zhu, 2016*). The yield-related traits, particularly TKW, BY, GY, and HI, also showed substantial improvement with the application of 30 mM GB and Pro. These findings suggest that GB and Pro not only enhance vegetative growth but also contribute to reproductive success under salt stress. The increase in TKW and GY can be attributed to improved photosynthetic efficiency and nutrient partitioning, as GB and Pro mitigate the detrimental effects of salt ions on chlorophyll content and enzymatic activities involved in grain filling (*Hasanuzzaman et al., 2019*).

Interestingly, the rise in average temperatures in 2022, particularly during critical growth stages like grain filling, may have introduced additional stress to rice plants beyond salt stress. Elevated temperatures during this period often reduce grain-filling duration and starch accumulation, negatively impacting grain weight and quality (*Ren et al., 2023a*). This climate-induced heat stress would likely exacerbate the physiological challenges posed by salt stress. However, the application of GB and Pro appears to have buffered these effects, as evidenced by the improvements in traits such as TKW and GY. The osmoprotectants likely enhanced the plants' resilience to both heat and salt stress by maintaining cellular functions and reducing oxidative damage (*Ren et al., 2023a*). Moreover, the increase in HI, particularly in plants treated with the combined GB and Pro

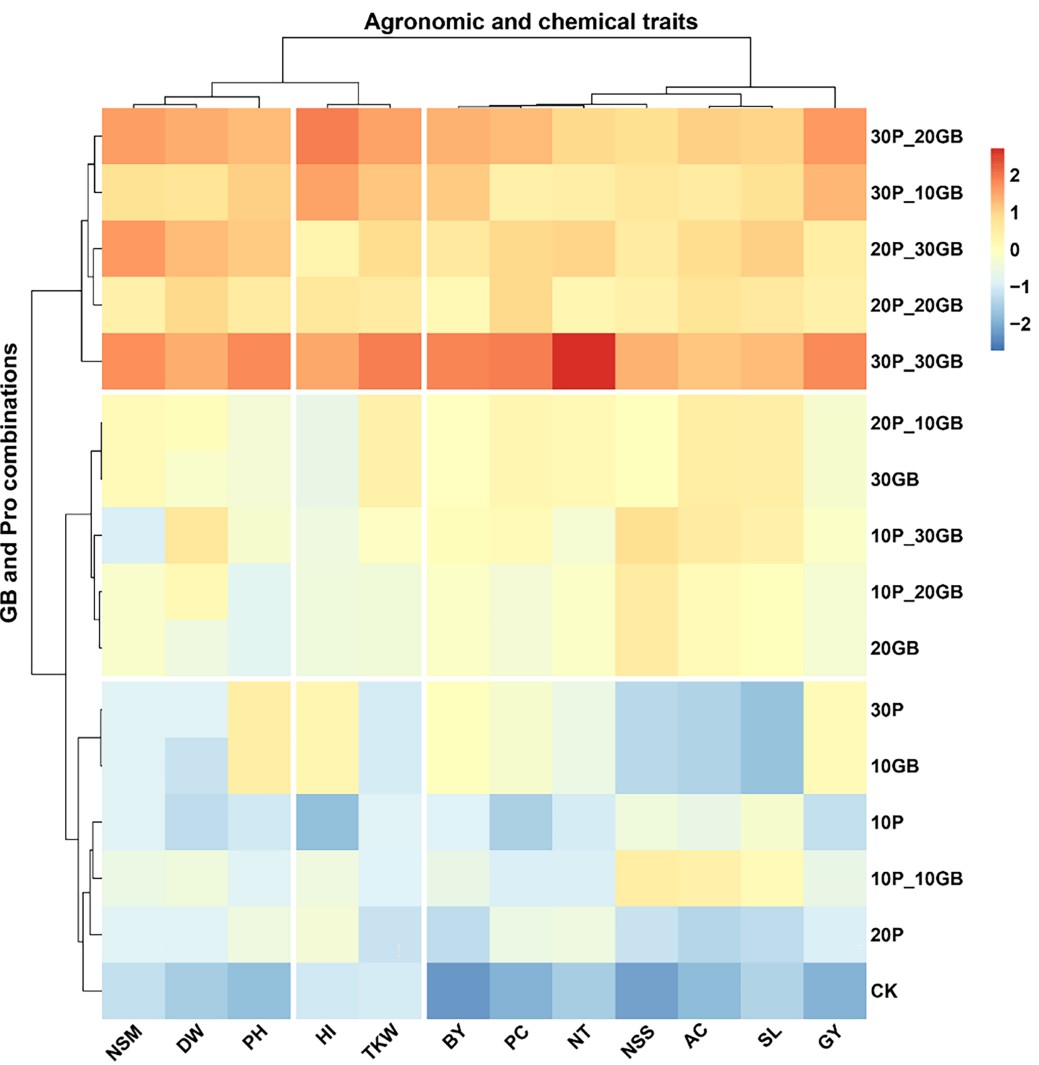

**Figure 4 Clustering heatmap illustrating the interrelationships between different combinations of glycine betaine (GB) and proline (Pro) treatments and 12 agronomic and biochemical traits of rice.** The heatmap shows three distinct clusters of traits, highlighting their differential responses to the various treatment combinations. Warmer colors indicate higher positive correlations, while cooler colors represent lower or negative correlations. Dry weight (DW), plant height (PH), tiller number (NT), number of spikes m2 (NSM), spike length (SL), number of spikelets per spike (NSS), 1,000-kernel weight (TKW), biological yield (BY), harvest index (HI), amylose content (AC), protein content (PC), and grain yield (GY).

treatment, highlights the improved resource use efficiency, where more biomass is converted into grain, reflecting the overall effectiveness of these treatments in enhancing rice productivity under stress (*Hasanuzzaman et al., 2019*).

The chemical attributes of the rice grains, including AC and PC, also showed significant improvement, particularly under the combined application of 30 mM GB and Pro could be likely due to the enhanced nitrogen assimilation and protein synthesis pathways activated by these osmoprotectants under stress conditions. The increased AC% suggests that GB and Pro may stabilize starch synthesis enzymes, promoting carbohydrate metabolism even

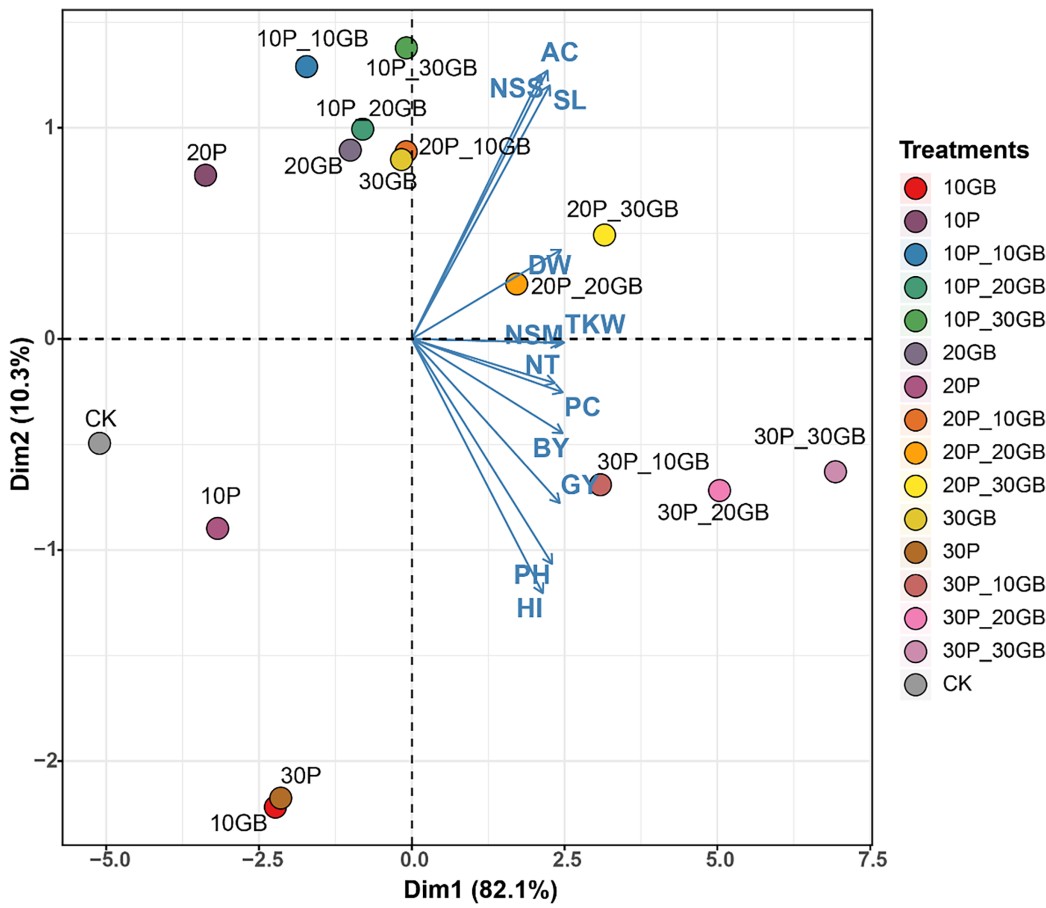

**Figure 5 Principal component analysis (PCA) biplot illustrating the combined effects of glycine betaine (GB) and proline (Pro) treatments on 12 rice traits.** The treatments include combinations of different GB and Pro levels, as well as the control (CK). The first principal component (Dim1) explains 82.1% of the variation, while the second component (Dim2) accounts for 10.3%. The arrows represent the direction and magnitude of each trait's contribution to the principal components. The GB and Pro treatment combinations are represented by colored dots, while traits are shown as vectors. Dry weight (DW), plant height (PH), tiller number (NT), number of spikes m2 (NSM), spike length (SL), number of spikelets per spike (NSS), 1,000-kernel weight (TKW), biological yield (BY), harvest index (HI), amylose content (AC), protein content (PC), and grain yield (GY).

under adverse conditions (*Hasanuzzaman et al., 2019*). The rise in protein content further points to maintaining of these compounds to N metabolism and reducing protein degradation caused by salt stress (*Iqbal et al., 2014*). This suggests that the combined application optimizes the osmotic balance, antioxidant defense mechanisms, and metabolic adjustments more effectively than individual treatments. Such synergy could be explained by the complementary roles of GB in osmotic adjustment and Pro in stabilizing subcellular structures and scavenging ROS, both of which are crucial for plant survival under stress (*Ren et al., 2023a*). Consequently, the combined use of GB and Pro represents a promising agronomic strategy for improving salt tolerance in rice, potentially enhancing crop resilience and productivity in salt-affected areas, which is increasingly important in the context of global climate change and soil salinization (*Roy, Negrão & Tester, 2014*).

The correlation analysis provides valuable insights into the interrelationships among various agronomic traits. The strong positive correlations observed between PH and traits such as GY, BY, HI, and others underscore PH's central role in crop performance, supporting findings that taller plants often exhibit enhanced photosynthetic capacity and biomass accumulation (*Ain et al., 2022*). PC exhibited significant correlations with key traits like DW, NT, and GY, indicating its potential as a marker for selecting high-yielding genotypes (*Chattopadhyay et al., 2019*). The correlations between AC and traits like thousand-kernel weight TKW and spike length SL highlight AC's role in determining grain quality and yield components (*Hu, Cong & Zhang, 2021*). Thousand-kernel weight's close association with multiple traits, including DW, PH, and GY, reaffirms its importance in plant productivity and yield (*Marone et al., 2021*). Similarly, spike length's correlation with traits such as DW and NSS emphasizes its influence on yield potential (*Siddiqui et al., 2017*). The robust correlations of HI with GY and BY underline its significance as a measure of resource-use efficiency, essential for achieving high yields.

The path analysis of agronomic and chemical traits in this study revealed significant insights into the factors influencing GY. The most impactful trait was TKW aligns with recent studies that highlight its critical role in yield enhancement (*Abdel-Aty et al., 2023*; *Jevtić et al., 2021*). Also, the significant positive direct effect of BY on GY emphasizes the importance of biomass accumulation in achieving higher yields, supported by its correlation with enhanced photosynthesis and grain filling (*Tian et al., 2022*). HI, which reflects the efficiency of biomass conversion to grain, significantly contributed to GY, aligning with current research that identifies HI as a key yield determinant in cereals (*AbdElgalil et al., 2023*; *Shaukat et al., 2024*; *Yang & Zhang, 2010*). Interestingly, NSS also had a positive effect, while SL had a significant negative impact on GY, suggesting a trade-off between spike length and yield due to resource allocation (*Xie & Sparkes, 2021*).

The most significant finding supplemented by clustering heatmap in this study is the strong synergistic effect observed when combining high concentrations of GB and Pro, particularly at the 30P_30GB level. This combination notably enhanced key traits such as PH, NSM, GY, and TKW, indicating that the dual application of these osmoprotectants effectively improves both vegetative growth and yield components. The synergy likely stems from the combined osmoprotective effects, which enhance cellular stability, osmotic balance, and stress tolerance, leading to improved overall plant performance under saline conditions. The weaker responses in treatments with only one osmoprotectant or lower concentrations underscore the necessity of using both GB and Pro together for optimal results. These findings are consistent with recent research highlighting the benefits of combined osmoprotectant application in enhancing stress resilience and crop productivity (*Abdelghany, Lamlom & Naser, 2024*; *Ren et al., 2023b*).

PCA of the combined effects of glycine GB and Pro on rice traits offers valuable insights into the interaction of these treatments with various growth parameters. The substantial variation captured by Dim1 (82.1%) in this study suggests that most of the observed differences among treatments can be attributed to this component, with treatments such as 30P_10GB, 30P_20GB, and 30P_30GB exhibiting similar positive impacts on key traits like NT, PC, BY, and GY. These findings are consistent with recent studies that highlight

the synergistic effects of osmoprotectants like GB and Pro in enhancing stress resilience and promoting growth under challenging environmental conditions (*Ghadirnezhad Shiade et al., 2023*). The distinct clustering of treatments along Dim2, accounting for 10.3% of the variation, further underscores the nuanced responses of traits such as DW, TKW, and NSM to different treatment combinations. The negative association of traits like PH and HI with Dim1 across most treatments suggests a differential response that could be indicative of trade-offs between growth and stress adaptation mechanisms (*Farouk et al., 2024*; *Nahar, Hasanuzzaman & Fujita, 2016*). The clear distinction of control and single-treatment groups from the combined treatment clusters highlights the enhanced efficacy of combined GB and Pro applications, as opposed to individual treatments, in modulating trait expression under salt stress conditions. This aligns with emerging evidence that the combined use of osmoprotectants can lead to more robust and consistent improvements in crop performance (*Yasmeen et al., 2024*).

## CONCLUSIONS

This study highlights the significant role of exogenous glycine betaine and proline in enhancing the growth, yield, and quality of rice (*Oryza sativa*) under salt stress conditions. Key findings include improved agronomic traits such as plant height, dry weight, and total nitrogen, alongside yield-related traits like thousand-kernel weight, grain yield, and harvest index. Notably, the combined application of GB and Pro (30P_30GB) demonstrated a synergistic effect, resulting in superior performance compared to individual treatments or lower concentrations.

Beyond addressing salinity, our study provides novel insights into the ability of GB and Pro to mitigate concurrent heat stress during the critical grain-filling stage, enhancing resilience and productivity under dual stress conditions. Additionally, improvements in grain quality traits, including amylose content and protein content, emphasize the potential of these osmoprotectants in sustaining both nutritional quality and yield in adverse environments.

The novelty of this study lies in demonstrating the dual efficacy of GB and Pro in mitigating multiple abiotic stresses, integrating advanced statistical methods such as principal component analysis to elucidate treatment-specific effects, and providing a comprehensive analysis of both agronomic and chemical traits. These findings present a promising agronomic strategy for improving the resilience and productivity of rice in saline and heat-affected regions, addressing global challenges related to climate change and soil salinization.

### Funding

This work was supported by King Saud University, project number (RSPD2025R931) Riyadh, Saudi Arabia. The funders had no role in study design, data collection and analysis, decision to publish, or preparation of the manuscript.

## Grant Disclosures

The following grant information was disclosed by the authors:
King Saud University, Riyadh, Saudi Arabia: RSPD2025R931.

## Competing Interests

The authors declare that they have no competing interests.

## Author Contributions

- Sobhi F. Lamlom conceived and designed the experiments, performed the experiments, analyzed the data, prepared figures and/or tables, and approved the final draft.
- Aly A. A. El-Banna conceived and designed the experiments, prepared figures and/or tables, authored or reviewed drafts of the article, and approved the final draft.
- Honglei Ren performed the experiments, prepared figures and/or tables, authored or reviewed drafts of the article, and approved the final draft.
- Bassant A. M. El-Yamany performed the experiments, prepared figures and/or tables, authored or reviewed drafts of the article, and approved the final draft.
- Ehab A. A. Salama performed the experiments, prepared figures and/or tables, authored or reviewed drafts of the article, and approved the final draft.
- Gawhara A. El-Sorady performed the experiments, analyzed the data, prepared figures and/or tables, authored or reviewed drafts of the article, and approved the final draft.
- Mohamed M. Kamara conceived and designed the experiments, prepared figures and/or tables, authored or reviewed drafts of the article, and approved the final draft.
- Amal Mohamed AlGarawi analyzed the data, prepared figures and/or tables, authored or reviewed drafts of the article, and approved the final draft.
- Ashraf Atef Hatamleh analyzed the data, prepared figures and/or tables, authored or reviewed drafts of the article, and approved the final draft.
- Abdelsalam A. Shehab analyzed the data, prepared figures and/or tables, authored or reviewed drafts of the article, and approved the final draft.
- Ahmed M. Abdelghany conceived and designed the experiments, performed the experiments, analyzed the data, prepared figures and/or tables, authored or reviewed drafts of the article, and approved the final draft.

## Data Availability

The raw data is available in the Supplemental Files.

## Supplemental Information

Supplemental information for this article can be found online at http://dx.doi.org/10.7717/peerj.18993#supplemental-information.

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
