# Peer review of "Synergistic effects of foliar applied glycine betaine and proline in enhancing rice yield and stress resilience under salinity conditions"

_PeerJ, doi:10.7717/peerj.18993_

## Round 0.1 · original submission · Major Revisions

Dear Dr. Lamlom

It is my opinion as the Academic Editor for your article - Synergistic effects of glycine betaine and proline in enhancing rice yield and stress resilience under salinity conditions - that it requires some revision.

The reviewers have highlighted a number of shortcomings in your manuscript, and there are a range of major and minor revisions in order to improve the quality and make your paper suitable for publication. You are therefore advised to go through reviewers' comments and suggestions, and modify your manuscript with focus on all the sections. Major emphasis should be on changes/modifications suggested and queries related to experimental design and procedures.

Please resubmit your revised manuscript for further assessment.

Reviewer 1 ·

Basic reporting

Dear Author,
I have carefully reviewed the article entitled “Synergistic effects of glycine betaine and proline in enhancing rice yield and stress resilience under salinity conditions”. Some major mistakes were found which needs to be rectified are mentioned below.
The abstract section needs to be reviewed carefully to ensure that the values and the years are mentioned accurately. Additionally, the keywords should be listed in alphabetical order for clarity.

Experimental design

In the introduction, the authors are advised to provide a declaration of the study's uniqueness in the conclusion, highlighting its distinct contribution. The general objective of the study should focus on the broader research aim, while the specific objective should detail the particular outcomes the study seeks to achieve.
This study has uncovered new research-related activities or relationships that were not identified in the previous study, demonstrating an expansion of the research gap.
The format of the reference should be same throughout the MS. In its current state, the level of English throughout the manuscript does not meet the journal's required standard. Also go through the captions of figure 2 present in the MS.

Validity of the findings

Kindly write the key findings of the study and the novelty in the conclusion section. The references are not according to the journal’s format. Kindly go through the journal and format accordingly. Also, italicize the scientific names in the reference part.

Additional comments

NA

Reviewer 2 ·

Basic reporting

Thank for giving the opportunity to review the article "Synergistic effects of glycine betaine and proline in enhancing rice yield and stress resilience under salinity conditions and review the MS and found that It need major revision before publication. Some of the major advise are here.
1. Check the values and year in the abstract portion.
2. What additional research related activities or relationships have been discovered in this
study as compared to the previous one?
3. In the introduction part, the authors should provide a declaration of the uniqueness of the study
4. Write the specific objectives of the current study.
5. What is the novelty of the current study?
6. From line no 103-105, reframe the sentences.
7. Kindly revise the line no. 122-126.
8. Improve all the captions words
9. In line no 189-201, add table number correctly after explaining the each effect of glycine
betaine and proline. As it is difficult for the readers to understand.
10. In line no 214-215, check the year in table 5 and mention correctly.
11. In correlation analysis of studied traits check the value for NSM and BY from figure 2.
12. Where are the values for AC mentioned in figure 2.
13. Check the footnote of all the figures mentioned.
14. The conclusion part is too small. Kindly summarize the key findings of the study along
with the novelty.
15. References need to go thoroughly again as some are not in format.
16 . material and methods must be more clear way
17 Some of the references not in proper format

Experimental design

Need some more clear

Validity of the findings

Fine more new references are needed

Additional comments

No

Reviewer 3 ·

Basic reporting

The article is written in good English.
The research background is not sufficient and there is a lack of information on salinity and yield relation in rice especially in Egypt.
What is the status of Rice cultivation in Egypt is missing in the introduction. How much area is covered under saline soils?
How much yield reduction is there due to salinity in Egypt?
What is the characters of variety Shakha 108?. Is it tolerant or susceptible?.
Hypothesis and research questions are not mentioned

Experimental design

research question and hypothesis is not clear and is missing completly
Material and methods are not complete and they are not reproducible.
eventhough two year study is perfomred the methodology is ambiguous and not complete
Line 99: year of study is 2020 and 2021 or 2021-22?
Clearly mention the characters of the variety used in the study
Line 103-104: The weather description of the location should be detailed with long-term average
Line 116: what is spray volume at different stages of spraying, and how many sprays are given
Line 136: is written that “was assumed”. The results cannot be assured. Change it “was recorded”.
What is the spacing of the crop
Which method of cultivation was adopted is it aerobic rice or submerged rice?
How the planting was done, direct-seeded or transplanted, what method of irrigation was followed, how many irrigations were given, and what was the irrigation water quality?
How the GB and proline solutions were prepared and what is the time of the day the spraying was carried out?
What is the duration of the experiment? What is the sowing date and harvesting dates?
Why split-plot design is used rather than factorial RBD.
Statistical analysis needs a relook.
The year effect is to be analyzed to see the impact of climate on crop productivity.
Since the experimental design is split plot, the main factor (glycine betaine) and sub-factor (Proline) means should be provided.

Validity of the findings

Line 169: please check the year is it 2021 and 2022 or 2020 and 2021 (mentioned in Tables)
Is there any received during the cropping season? It is not shown in figure 1
Line 180-182: mentioning the weather parameters during the study doesn’t carry any meaning, it should be correlated with yield and other traits in relation to weather.
Authours are requested to do the biplot analysis of weather and yield and yield related traits to see the influence of year effect
Line 191 and 194, 198: it mentioned that “compared with those in the control treatment”, but there is no comparison of treatment was made with control. Kindly rewrite or correct the statement.
Line 203-205: in the second season DW in 30 mM GB and 30 mM Pro is on par with 20 mM GB and 30 mM Pro also. So authours need to mention this also.
231-239: figure 2 it was observed that all the parameters are positively correlated with each other except NSS with HI. But in Figure 3 path analysis some are negatively contributing. How justify
Line 236: SL is it Spike length of the Number of spikelets please check and rewrite.
Control for proline is missing in Table 2
Figure 2 needs to be restructured. Keep the main dependent variable at the top and the other following the dependent variable. So that reading the image is easier.
Tillers recorded in the study are a number of productive tillers or all tillers.
It is strange to see the 1000 kernel weight the same during both years (Table 3)
10GB-30Pro also recorded at par yield 5.24 and 5.63 during 2020 and 2021 compared to 20GB-30Pro and 30GB-30Pro.
Just because of the increase in 10µM Pro there is an increment of 2t grain yield. The results are very encouraging and what might be the mechanism to enhance yield by aprox. 30%.
280-281: there is no clear analysis of the impact of weather on various rice morphological parameters. Therefore, the statement made in line 280-281 is just hypothetical. Proper statistical analysis between weather and traits is required to prove the relation and impact between them.
For most of the improvement in yield and its traits authors mentioned that this may be due to improvement in many physiological functions such as osmotic potential, turgor pressure, and photosynthesis. All these statements are just hypothetical and there is no evidence to prove the effect of GB and Proline except from literature.
In conclusion, authours suggesting 30GB-30P as the best treatment, however, there is at par result with 20GB-30P and 10GB-30P. what is the reason behind suggesting 30GB-30P

Additional comments

A study of the effect of osmo protectants on rice in saline soils is a good study. However, the background of the research is not clear. There are many details and information on basic aspects of crop and research problems that are missing in the introduction part. The methodology is not complete and many details are not mentioned. Detailed comments are given for further improvement.
How do the authours justify that there was a salinity stress? There is no control plot or osmotic potential was measured.
A salinity of 2.6 dSm-1 is less than the threshold limit of salinity i.e., 4dSm-1. Rice can tolerate this salinity. The variety used may be susceptible to this salinity level?. What is the yield level of this variety in non-saline soils and how much yield reduction due to salinity needs to be mentioned in the introduction? Pooled analysis of the year required to be done and the way of data presentation needs to be changed including year variability.
figure 2 it was observed that all the parameters are positively correlated with each other except NSS with HI. But in Figure 3 path analysis some are negatively contributing. Looks like there is a problem with the analysis or data.
For most of the improvement in yield and its traits, authors mentioned that this may be due to improvement in many physiological functions such as osmotic potential, turgor pressure, and photosynthesis. All these statements are just hypothetical and there is no evidence to prove the effect of GB and Proline except from literature.

·

Basic reporting

Comments in file attached

Experimental design

Comments in file attached

Validity of the findings

Comments in file attached

Additional comments

Comments in file attached

---

## Round 0.2 · Minor Revisions

Dear Dr. Lamlom,

Thank you for your submission to PeerJ.

It is my opinion as the Academic Editor for your article - Synergistic effects of foliar applied glycine betaine and proline in enhancing rice yield and stress resilience under salinity conditions - that it requires a few more Minor Revisions.

Reviewer 3 ·

Basic reporting

well written

Experimental design

Well written and still few calrifications required
• What is the characters of variety Shakha 108?. Is it tolerant or susceptible?.
Response: It is known for its high yield potential and relatively good adaptability to varying environmental conditions. Shakha 108 is generally considered to be susceptible to salt stress, especially when compared to some salt-tolerant varieties.
New Comment: Mention these charters in manuscript
• Line 116: what is spray volume at different stages of spraying, and how many sprays are given.
Response: The application of glycine betaine (GB) at concentrations of 0 mM, 10 mM, 20 mM, and 30 mM was carried out via foliar spraying throughout three rounds at 30, 45, and 60 days after sowing (DAS). Similarly, the subplots were sprayed with four different concentrations of proline (Pro) (0 mM, 10 mM, 20 mM, and 30 mM) during the same periods
New Comment It was asked to give spray volume used for each treatment not concentration of osmoprotectants. Kindly mention in methodology about how much volume of spray solution is used during different stages
• Comment: What is the duration of the experiment? What is the sowing date and harvesting dates?
Response: We add it in Lines 161-163
New Comment No information was mentioned about harvesting date or after how many days crop was harvested.
• How the GB and proline solutions were prepared and what is the time of the day the spraying was carried out?
Response: The glycine betaine (GB) and proline solutions were prepared by dissolving each compound in distilled water to achieve the desired concentrations, as specified in the experimental design. The spraying of GB and proline was conducted at early morning hours (typically between 7 a.m. and 9 a.m.) to minimize evaporation and enhance absorption, allowing the plants adequate time to take up the solutions before the midday sun intensified. This timing also helped reduce the risk of phototoxicity and ensured optimal effectiveness of the treatments.
New Comment Still this information in methodology is not mentioned in revised material and methods. Mention this detailed procedure in material methods
• Since the experimental design is split plot, the main factor (glycine betaine) and sub-factor (Proline) means should be provided.
Response: Thank you for your suggestion. We appreciate your attention to the experimental design and the need to clearly present the results of the main factor (glycine betaine) and the sub-factor (proline) in the analysis. As the study follows a split-plot design, the main factor (glycine betaine, GB) was applied at the plot level, while the sub-factor (proline, Pro) was applied at the subplot level.
New Comment It was asked to give mean values for main plot and subplot treatments in the Tables.

Validity of the findings

conclusions are well stated as per the objectives, justifications are appropriate for the findings

Additional comments

The authours addressed most of the comments and few are need to address provoided for inclusion

---

## Round 0.3 · accepted · Accept

Dear Dr. Lamlom

Thank you for your submission to PeerJ.

I am writing to inform you that your manuscript - Synergistic effects of foliar applied glycine betaine and proline in enhancing rice yield and stress resilience under salinity conditions - has been Accepted for publication.

Congratulations!


Reviewer 3 ·

Basic reporting

manuscript is well structured with relavent figures and tables, language is clear.

Experimental design

methodology is suffiicnet and regiorous results provided to make results technically strong

Validity of the findings

clear and well written with conclusion

Additional comments

manuscript is well written and all comments are addressed